# Longing for Touch and Quality of Life during the COVID-19 Pandemic

**DOI:** 10.3390/ijerph20053855

**Published:** 2023-02-21

**Authors:** Birgit Hasenack, Larissa L. Meijer, Jonas C. C. Kamps, Andy Mahon, Giulia Titone, H. Chris Dijkerman, Anouk Keizer

**Affiliations:** 1Experimental Psychology, Faculty of Social and Behavioural Sciences, Utrecht University, 3584 CS Utrecht, The Netherlands; 2Clinical Psychology, Faculty of Social and Behavioural Sciences, Utrecht University, 3584 CS Utrecht, The Netherlands

**Keywords:** longing for touch, quality of life, touch deprivation, COVID-19

## Abstract

To combat the spread of the COVID-19, regulations were introduced to limit physical interactions. This could induce a longing for touch in the general population and subsequently impact social, psychological, physical and environmental quality of life (QoL). The aim of this study was to investigate the potential association between COVID-19 regulations, longing for touch and QoL. A total of 1978 participants from different countries completed an online survey, including questions about their general wellbeing and the desire to be touched. In our sample, 83% of participants reported a longing for touch. Longing for touch was subsequently associated with a lower physical, psychological and social QoL. No association was found with environmental QoL. These findings highlight the importance of touch for QoL and suggest that the COVID-19 regulations have concurrent negative consequences for the wellbeing of the general population.

## 1. Introduction

It has repeatedly been shown that positive interpersonal touch has beneficial effects on social, physical and psychological wellbeing [1]. For example, touch plays an essential role in bonding and the maintenance of social relationships [2]. Touch also reduces short-term stress levels by lowering blood pressure and heart rate [3,4]. This has, in turn, been associated with improved immune system functioning [5]. Receiving touch also impacts psychological wellbeing: massages have been found to have a positive impact on depression [6], eating disorders [7] and post-traumatic stress disorder [8]. Taken together, these studies highlight the importance of frequently receiving touch.

To contain the spread of the COVID-19, public health measures such as social distancing, (self-) isolation and quarantine were implemented, and are still present in some affected countries [9]. These public health measures have included the prohibition of social gatherings and the advice or mandate to stay at home as much as possible. Although these regulations have effectively slowed the transmission of the virus [10], recent research shows that they are also associated with an increase in longing for touch [11,12]. Longing for touch is defined as a significant discrepancy between touch frequency and touch wish, and refers to social consensual touch [13]. Given the aforementioned importance of touch, a rise in longing for touch could subsequently have a severe negative impact on our physical and psychological wellbeing, and our quality of life (QoL).

To our knowledge, no study to date has examined the potential relation between longing for touch and QoL. Pre-pandemic studies about the effects of touch deprivation have primarily been conducted in children [5,14]. Therefore, less is known about the consequences of longing for touch for (healthy) adults and, in particular, about the effects of moderate forms of deprivation [14]. An exception is a study by Floyd (2014), which shows associations between affection deprivation (i.e., longing for affectionate touch) in adults and various indices of social, physical and psychological wellbeing, such as loneliness, stress and general health [1]. However, this study did not conduct a systematic analysis of the relation between affection deprivation and QoL. In addition, participants in this study reported a longing for touch while living without government-imposed social restrictions. It is, therefore, unclear whether these results can be generalized to individuals who live in a society in which such restrictions are enforced.

Although several studies have investigated the effects of the COVID-19 pandemic on QoL [15,16,17,18,19], none of these studies have considered the potential relation with longing for touch. These studies have, however, demonstrated that the pandemic is associated with a decrease in QoL. Most of these studies have focused on health-related correlates of QoL. For example, lower health-related QoL has been observed in China [15,16] and Austria [17]. Lower psychological, physical, social and environmental QoL has been reported in Greece [18] and Italy [19]. In addition, Field et al. (2020) reported that touch deprivation during the pandemic is associated with increased stress, depression symptoms, fatigue and sleep quality [20]. Furthermore, COVID-19-related touch deprivation has also been associated with increased anxiety and more loneliness [12]. Although these findings emphasize the importance of touch, direct measures of QoL are not included in these studies.

The aim of the current study is, therefore, to investigate the association between longing for touch, QoL and COVID-19-related social restrictions. The relationship between longing for touch and COVID-19 social regulations has already been extensively discussed in a recent study by Meijer et al. (2022), and is, therefore, beyond the scope of the current study. Firstly, the insights of the current study could be considered during future pandemics or situations in which nationwide social restrictions have to be imposed. If there is an association between longing for touch and QoL, potential interventions could be developed to prevent or reduce touch deprivation while taking the necessary regulations into consideration. We have previously demonstrated that longing for touch was highly prevalent during the pandemic [11]. As such, it is important to understand which correlates of wellbeing are associated with this. Second, longing for touch is not exclusively linked to the COVID-19 pandemic. This is supported by a study of Beßler et al. (2020), who also observed a relatively high prevalence of longing for touch in a pre-pandemic setting [13]. Therefore, investigating the relation between longing for touch and QoL will not only have potential implications for future pandemics, but also for those who feel touch-deprived or long to be touched in general. Based on previous research, we expect longing for touch to be negatively associated with social, physical and psychological QoL [14]. Furthermore, similar to previous studies [17,18,19] we expect a negative relation between COVID-19 regulations and QoL. Specifically, we expect that a more severe regulation (i.e., a lockdown or complete lockdown) will have a negative influence on QoL. We also expect that the duration of these regulations will be negatively associated with QoL, meaning that scores will decrease when regulations are implemented for a longer period of time.

## 2. Methods

### 2.1. Participants

The data presented in this paper are a subset derived from a larger online survey that was administered to participants in different countries (see Meijer et al., 2022, as well [11]). Participants were recruited through social media posts that were shared by the authors and several national/regional media platforms. A total of 2403 participants completed the survey, but for the purpose of this study, only those who were currently experiencing COVID-19 regulations at the time of testing were included. Participants also needed to be older than 16 and not have been diagnosed with neurological, mental or skin disorders. This resulted in the exclusion of 373 participants. An additional 48 participants with anomalous scores for the duration of regulations were excluded from subsequent analyses (+3 SD), which meant that the final sample size was 1982. The majority of these participants were female (*n* = 1578) and the age ranged between 16 and 87 (*M* = 38.53, *SD* = 15.61). For a detailed description of sample demographics, see Appendix. All participants provided written consent before the start of the experiment and did not receive any form of compensation. The study was approved by the local faculty ethical review board of Utrecht University (protocol number 20-210) in accordance with the Declaration of Helsinki.

### 2.2. Materials

#### 2.2.1. Longing for Touch Questionnaire

Before the questionnaire, participants were informed about the type of touch referred to in this study through the information letter. Here, it was described that the study focused on social touch. The Longing for Touch Questionnaire contained two items (“*Currently I would prefer to be touched by others* …” and “*Currently I would prefer to touch others* …”). Participants used a VAS to answer these questions, with responses ranging from 0 (“*Currently I would prefer to be touched less by others/to touch others less*”) to 10 (“*Currently I would prefer to be touched more by others/to touch others more*”). An average longing for touch score was subsequently calculated by taking the mean across the two items, with scores above 5 being indicative of a longing for touch. The reliability of this questionnaire was high, Cronbach’s α = 0.92. This questionnaire has been used in our previous study [11] (Meijer et al., 2022).

#### 2.2.2. WHOQOL-BREF

The World Health Organization Quality of Life Instrument (WHOQOL-BREF) was used to measure QoL. This questionnaire is a shortened version of the WHOQOL-100 and comprises 26 questions that are combined in four different domains (physical health, psychological health, social relationships, environment). Q1 (“How would you rate your quality of life?”) and Q2 (“How satisfied are you with your health?”) are analysed separately from the domains. Responses to all questions were given on a 5-point Likert scale, ranging from 0 (“Not Satisfied”) to 5 (“Very Satisfied”). The domain scores were subsequently transformed and computed to conform to WHO guidelines [21]. In line with these guidelines, scores were multiplied by 4 to allow them to be compared to scores on the WHOQOL-100 [21]. For all domains, higher scores reflected a higher QoL.

#### 2.2.3. COVID-19 Regulations

Information about the duration and severity of current COVID-19 related regulations was obtained at the start of the study. Participants were asked to identify which regulations were in place during the time of participation. The regulations were divided into four subcategories: (1) the advice not to shake hands (ANSH), (2) social distancing, (3) lockdown and (4) complete lockdown (CL). These categories were based on the regulations in Europe at time of designing the study. The advice not to shake hands was indicated as the least severe and meant that people were advised by the government to not shake hands or to avoid touching each other. Social distancing meant that people were obligated to keep 1.5 m away from one another and asked to disengage from and avoid any social interactions. Lockdown was defined as the advice to stay at home unless you need to go out, with imposed fines for those who do not adhere to regulations. Finally, complete lockdown was deemed to be the most severe regulation and was defined as it being prohibited to leave the house without a clear purpose and, when leaving, it being compulsory to remain within a close radius of one’s own house.

### 2.3. Procedure

Data were collected with an online survey on Qualtrics between 5 April and 8 October 2020 (see Meijer et al., 2022 for more information [11]). All participants provided informed consent at the start of the experiment. Participants first completed a number of demographic questions and then indicated if there were any COVID regulations in place, how long these had been in place and how severe these regulations were. They subsequently filled out the Longing for Touch Questionnaire and the WHOQOL-BREF. The order in which these questionnaires were filled out was the same for each participant. Participants were not debriefed at the end of the experiment, but were provided with a link to a documentary about touch deprivation if they were interested in learning more about the topic of the study.

### 2.4. Data Analysis

All analyses were executed in SPSS 24.0. (SPSS Inc., Chicago, IL, USA) Multiple linear regressions with bias corrected accelerated (BCa; 1000 iterations) were used for the analysis. Data were checked for multicollinearity (VIF < 5). The variables were added to the regression model with forced entry. Unless stated otherwise, α = 0.01.

## 3. Results

### 3.1. Demographics

Most participants lived in the Netherlands (68.1%) or Italy (11.2%) (also see Table A1 in the Appendix A). At the time of the study, the majority of participants were in lockdown (62.2%). These public health measures had been in place between 0 to 130 days (*M* = 42.82, *SD* = 23.93). A total of 83.01% of participants had a longing for touch score of 5 or higher (*M* = 7.7, *SD* = 2.3). Note: the percentage reported here differs slightly from Meijer et al. (2022) [11] because the sample size is smaller due to exclusion criteria of the WHOQOL-BREF.

### 3.2. Quality of Life

In line with WHO guidelines, participants who answered less than 20% of the questions were excluded from this analysis [21]. This meant that four participants were not used in the subsequent analyses (*n* = 1978). Multiple linear regressions were conducted on each of the four domains. Between-subject factors were the duration and severity of social distancing regulations, and longing for touch. Descriptive statistics for each domain are displayed in Table 1.

The regression coefficients for physical, psychological, social and environmental QoL are displayed in Table 2. First, the model significantly predicted physical QoL F(5, 1867) = 9.81, *p* < 0.001, R^2^ = 0.03. A higher longing for touch was significantly associated with a lower physical QoL. Participants in a complete lockdown also experienced a significantly lower physical QoL than those in a lockdown and participants who were social distancing. No difference was observed between advice not to shake hands and a complete lockdown.

Second, the model significantly predicted psychological QoL, F(5, 1869) = 5.62, *p* < 0.001, R^2^ = 0.12. A higher longing for touch was significantly associated with a lower psychological QoL. Participants in a complete lockdown also had a significantly lower psychological QoL than those in lockdown and those who were social distancing. There was no significant difference between advice not to shake hands and a complete lockdown. Third, social QoL was significantly predicted by the model, F(5, 1870) = 20.71, *p* < 0.001, R^2^ = 0.23. Both longing for touch and the duration of the regulations were significantly negatively associated with social QoL. No significant effect was found for the severity of the regulations. Fourth, environmental QoL was significantly predicted by the model, F(5, 1870) = 22.92, *p* < 0.001, R^2^ = 0.06, but neither longing for touch nor duration of regulations had a significant effect. However, the severity of regulations was a significant predictor; participants in complete lockdown scored significantly lower on environmental QoL than those who had the advice not to shake hands, who were social distancing or were in a lockdown.

## 4. Discussion

Touch is important for social bonding and maintaining social relationships [2]. Furthermore, it has been shown that touch can reduce levels of stress and pain [3,22]. It is, therefore, crucial to receive touch on a regular basis. During the COVID-19 pandemic, social distancing regulations were implemented, and these had a severe impact on the ability to engage with one another and importantly to receive or provide touch. Touch deprivation has previously been linked to a reduced general wellbeing [17,18,19]^,^. The aim of the current study is, therefore, to investigate whether there is an association between longing for touch during the COVID-19 pandemic and the four domains of QoL.

In our sample, 83% of the participants experienced a longing for touch (see Meijer et al., 2022 for a thorough discussion of these results [11]). Consistent with our expectations, higher levels of longing for touch are associated with a lower physical, psychological and social QoL. No associations are found with environmental QoL. Since questions in this domain mainly relate to the participant’s financial situation, their physical environment and access to, among others, health services and transport, it seems unlikely that that these facets of QoL are directly influenced by touch and lack thereof.

To our knowledge, this is the first study demonstrating the relation between longing for touch and physical, psychological and social QoL in a large community sample. These results show that being touch deprived might have a severe impact on our general wellbeing. Previous research shows that touch has several beneficial effects on wellbeing, which can be linked to the different QoL domains. First, the physical QoL domain pertains to, among others, pain, energy and sleep quality. Touch before sleep is suggested to have a positive effect on sleep quality [23]. Furthermore, a recent study [20] shows that touch deprivation is related to poorer sleep quality. Second, the items in the psychological QoL domain relate to feelings of happiness, self-esteem and the presence of anxiety, low mood and depression. Previous studies have shown that touch can reduce stress, anxiety and feelings of depression [2,3,6]. Third, the social QoL domain contains questions related to the quality of personal relationships and the perceived level of social support. Touch can promote social bonding [2], is associated with social support [1] and can reduce feelings of social exclusion [24]. These previous studies show the importance of touch in daily life and its positive influence on general wellbeing. This is the first study with a large community sample to show that touch not only has a positive influence on our wellbeing, but that when there is a lack of touch, this has negative consequences, which further emphasizes the importance of touch for QoL in general.

It should be mentioned, however, that the used model only explained a relatively small portion of the variance in these domains. This is perhaps not surprising, given that a large variety of factors can have an influence on QoL, such as age, gender, social functioning, education, employment and self-efficacy [25]. One could also argue that, on the other hand, the found effect sizes might indicate that longing for touch has a relatively large influence on QoL, given that it depends on a large number of factors and we only included three predictors in the model. In addition, the design of the current study also does not allow for drawing causal conclusions. A lack of social relations and support could equally lead to an increased longing for touch. Similarly, problems with physical and psychological wellbeing could lead to (social) isolation and subsequently contribute to a longing for touch. However, these results do demonstrate that there is an association between longing for touch and various domains of QoL.

On the other hand, it could also be that participants with lower QoL have higher levels of longing for touch. QoL could have been lowered during the pandemic by other factors, such as (un)employment and the ability to go to work. This, in turn, could lead to a desire to receive or provide more touch. Since previous research shows that there is a positive relationship between receiving touch and general wellbeing [6,23,24], it is expected that longing for touch influenced QoL in the current study rather than vice versa. However, the current design does not allow us to draw any direct conclusions about the causality of this relation.

As well as longing for touch, the current study also investigated the association between the duration and severity of the COVID-19 regulations and QoL. The duration of the regulations was only associated with social QoL. Interestingly, there was no relation between the severity of the regulations and scores on this domain. This seems to suggest that the duration of social restrictions plays a more important role than the type of restriction itself. In contrast, the severity of regulations did significantly predict scores in the physical, psychological and environmental QoL domains. The effect seems to be the largest for the environmental QoL domain. Participants who were in a complete lockdown consistently score lowest on each of these domains. These findings are consistent with the observation that COVID-19 regulations have had a strong impact on various societal aspects that are assessed within these domains, such as the inability to move around freely, exercise and go to work, increased anxiety and stress, and potential limited access to general healthcare [18,19]. A complete lockdown might also indirectly reflect the severity of the pandemic, which might be related to a decrease in a variety of health-related parameters, such as a higher chance of contracting the virus [17,18,19]. For physical and psychological QoL, no difference was observed between participants who received the advice not to shake hands and participants in a complete lockdown. However, this might be explained by the unbalanced number of participants in each group. The advice not to shake hands group only contains 34 participants, compared to 312 in the complete lockdown group. As mentioned, this is, to our knowledge, the first study in which a large community sample is used to study the association between longing for touch and the different QoL domains during the COVID-19 pandemic. Furthermore, as the COVID-19 pandemic and the social restrictions affected every person in society, this is a unique community-wide sample. Our results show that longing for touch and its influence on wellbeing is not related to a specific group, but is present in every layer of the society. However, it should be noted that although participants were recruited via a variety of social media platforms and all participants self-selected their participation, some groups, such as women, seem to be overrepresented. This might be because they have more affinity with the topic of our study. Future research should, therefore, aim to further eliminate selection bias by actively recruiting individuals who appear to be underrepresented in our current sample, such as men.

Our results show that the COVID-19 regulations are associated with QoL and, therefore, with societies’ wellbeing. To fully understand and interpret the impact of these regulations, it is important to compare the scores in our study with pre-pandemic data. A direct comparison is not possible, given that we do not have data about QoL in the current sample collected before the pandemic. However, as our data are based on a primarily Dutch sample, an indirect comparison can be made with the findings of Schrier and colleagues (2016), who investigated QoL in the general Dutch population and in psychiatric outpatient groups [23]. They report the following average scores for physical QoL (15.5), psychological QoL (14.7), social QoL (15.2) and environmental QoL (15.9). The psychiatric outpatients were reported to have significantly lower scores in all domains: physical QoL (11.8), psychological QoL (10.5), social QoL (12.8) and environmental QoL (13.5). Compared to these pre-pandemic norms in the general Dutch population, the current results show a lower average in all the four domains: physical QoL (14.8), psychological QoL (13.6), social QoL (13.0) and environmental QoL (15.3), also see Table 1. Importantly, social QoL scores during the COVID-19 pandemic are almost similar to the pre-pandemic social QoL of psychiatric outpatients. This further highlights the possible influence the COVID-19 social regulations have on social QoL.

The current study provides important insight into the association between COVID-19 related social restrictions, longing for touch and QoL. In particular, the influence on social QoL compared to pre-pandemic norms shows that a nationwide pandemic can have severe consequences on our general wellbeing. Therefore, if social restrictions are necessary to protect society from a virus like COVID-19 in the future, the influence on QoL and mental health should be taken into account. As it seems that longing for touch is associated with QoL as well, providing touch interventions might have a positive influence on QoL during a pandemic. As, of course, it is difficult or sometimes even impossible to touch another human during a pandemic with social restrictions, touching animals might have a positive influence on touch deprivation and, therefore, on our mental health. Recent research shows that touching and petting animals can also provide feelings of social safety and reduce stress and feelings of depression [26].

## 5. Conclusions

We investigated the relation between longing for touch, COVID-19-related public health measures and QoL. Longing for touch is associated with lower Social, physical and psychological QoL, of which the strongest association is with social QoL. Furthermore, the duration of the COVID-19 regulations is related to lower social QoL. In contrast, the severity of these regulations is only related to physical, psychological and environmental QoL. These findings provide further insight into the effect of COVID-19 public health measures and particularly the inability to frequently perceive and provide touch on QoL and wellbeing of the general population. Our findings emphasize the importance of touch for social, physical and Psychological QoL.

## Figures and Tables

**Table 1 ijerph-20-03855-t001:** Descriptives.

	Mean	SD	Range
Physical QoL	14.79	2.39	6–20
Psychological QoL	13.56	2.44	5–20
Social QoL	13.03	3.35	4–20
Environmental QoL	15.29	2.12	7–20

**Table 2 ijerph-20-03855-t002:** Regression coefficients.

Model	Physical QoL	Psychological QoL	Social QoL	Environmental QoL
Duration of regulations	−0.05 (−0.01, 0.00)	−0.03 (−0.01, 0.00)	−0.14(−0.03, −0.01) **	−0.06 (−0.01, −0.00)
Regulation severity	-	-	-	-
ANSH vs. CL	0.04 (−0.16, 1.68)	0.06 (0.19, 2.07)	0.05 (−0.13, 2.64)	0.11 (0.96, 2.59) **
Social distancing vs. CL	0.18 (0.69, 1.45) **	0.13 (0.39, 1.20) **	0.07 (−0.02, 1.11)	0.30 (1.20, 1.94) **
Lockdown vs. CL	0.16 (0.46, 1.07) **	0.12 (0.26, 0.92) *	0.03 (−0.23, 0.64)	0.32 (1.09, 1.65) **
Longing for touch	−0.11(−0.16, −0.06) **	−0.08 (−0.13, −0.03) *	−0.17 (−0.32, −0.18) **	0.02 (−0.02, 0.06)

Note: Multiple linear regression model with standardized betas (95% bootstrap corrected CI reported in parentheses). ** <0.001, * <0.01. ANSH = advise not to shake hands, social distancing = keeping at least 1.5 m away from each other, lockdown = the advice to stay at home unless you need to go out, CL = complete lockdown defined as it being prohibited to leave the house without a clear purpose and, when leaving, it being compulsory to stay within a close radius of one’s own house.

## Data Availability

The data presented in this study are available on request from the corresponding author. The data are not publicly available due to ethical reasons; sharing data in a publicly accessible repository was not included in the informed consent form.

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
