# Peer review of "Longing for Touch and Quality of Life during the COVID-19 Pandemic"

_ijerph, 2023, doi:10.3390/ijerph20053855_

Round 1

Reviewer 1 Report

Thank you for the opportunity to review this paper.  It makes a number of interesting points.  However, there are a number of conceptual and methodological issues which I feel impacts its potential for publication.

1) the measure and the paper itself does not provide an overview of what it meant by touch - was guidance given on this?

2) given the cultural implications of touch, and the variation in COVID responses, breaking respondents down by continent it not entirely helpful. I'm also unsure what benefit using such a small number of non-European responses brings.

3) were demographic details not included in the model?  I would think that age, gender, nation would all have an impact here which could be explored?

4) statements are made about restrictions and QoL throughout, but it is unclear whether this was the focus of your questionnaire. You claim the restrictions were impacting QoL, but were there not other factors connected with the pandemic which impacted this also?

5) much of the analysis is hindered by a lack of a control group - how can we rigorously know that longing for touch changed, or QoL, in this time period?

Line 21 - apologies if this a bit abstract, but I think there needs to be a definition of what you mean by longing to touch. Without this, any physical contact (e.g. abuse) could be included in your statements on the benefits.

Line 70-72 - while I agree with the statement here, I wonder if a caveat is also required to state that a similar pandemic may mean that subsequent restrictions are necessary. Just to avoid a potential misintreptation that a longing for touch would supersede the importance of similar restrictions.

Line 93 - typo of dan / than.

Line 88 - more discussion of how the questionnaire was circulated is required.  You note the very high percentage of female respondents, so understanding which groups were targeted, and how, would help an appraisal of this.

Line 102 - given the limited previous publications using this measure, and the international audience, was more information given to participants?  The simplicity of being asked whether or not I would prefer (relative to what, in what context?) to be touched (what form of touch?) could lead to significant variation in response.

Line 128 - I feel the use of the word "strict" here is leading.  Can this be justified?

Line 142 - more deomgraphic data is needed.  Which other nations were included?  Is there any consideration to the cultural significance given to touch in each participating culture?  

Line 167 - this data brings challenges without a comparison group of levels pre-pandemic (e.g. a baseline?).  Is there not an argument that a longing for touch could have an impact on QoL without any restrictions in place?

Line 192 - again, is this significant?  Is this a high or low level?

Line 216 - this is a good point, so were any of these included in your analysis / study design?

Line 235 - I feel that restrictions as the main focus are problematic at times here - we know that, in general, the pandemic had an impact on QoL in many different ways, not just the restrictions.  Was this addressed in the questionnaire?  Were people asked to reflect whether their QoL was primarily impacted by the restrictions or the pandemic itself?

Line 264 - again, why just the impact of restrictions?  Were other factors not impacting on QoL during this period?

Line 301 - given the nature of your sample, with 91% returning data from Europe, I think a discussion of what the international data brings is required.  Also, given the variation of responses to COVID by individual nations, grouping the findings by continent is not entirely helpful.

Reviewer 2 Report

This study examined the associations among Covid restriction policies, longing for touch and quality of life among a Dutch and Italian sample. The findings showed that stricter restrictions lowered QoL indices, and longing for touch could negatively predict the QoL. The study explored the role of interpersonal touching during a period that was marked by the scarcity of it, and has intervention implications. However, the study also has many shortcomings. 

1.     Since the authors put a lot of emphasis on the importance of longing for touch in the introduction, it is rather strange that they did not examine how covid restriction affects participants’ longing for touch. Is it that stricter restriction could predict higher longing for touch? And, is it possible that the restrictions affect QoL partially through the increased longing for touch?

2.     As a key predictor in the regression, the covid restriction types need more detailed explanation. How are they defined, how are they measured, their differences, hypothesized influences on QoL, etc. 

3.     Notes below the regression table should briefly explain the restriction types.

4.     In the current theoretical framework, it could be said that the research findings merely confirmed past research that longing for touch negatively affects quality of life; the theoretical advances should be spelled out more clearly.

5.     Grammatical error. “Conform WHO-guidelines, participants….” Line 150.

6.     The possibility that low QoL elicits a greater longing for touch cannot be ruled out; which merits a discussion.

Round 2

Reviewer 2 Report

The authors have made adequate revision, and the manuscript is clearer now.  I have no further comment.